# Individual DNA Methylation Pattern Shifts in Nanoparticles-Exposed Workers Analyzed in Four Consecutive Years

**DOI:** 10.3390/ijms22157834

**Published:** 2021-07-22

**Authors:** Andrea Rossnerova, Katerina Honkova, Irena Chvojkova, Daniela Pelclova, Vladimir Zdimal, Jaroslav A. Hubacek, Lucie Lischkova, Stepanka Vlckova, Jakub Ondracek, Stepanka Dvorackova, Jan Topinka, Pavel Rossner

**Affiliations:** 1Department of Genetic Toxicology and Epigenetics, Institute of Experimental Medicine CAS, Videnska 1083, 142 20 Prague 4, Czech Republic; katerina.honkova@iem.cas.cz (K.H.); irena.chvojkova@iem.cas.cz (I.C.); jan.topinka@iem.cas.cz (J.T.); 2Department of Occupational Medicine, First Faculty of Medicine, Charles University in Prague and General University Hospital in Prague, Na Bojisti 1, 120 00 Prague 2, Czech Republic; daniela@pelclova.cz (D.P.); Lucie.Lischkova@vfn.cz (L.L.); stepanka.vlckova@vfn.cz (S.V.); 3Department of Aerosol Chemistry and Physics, Institute of Chemical Process Fundamentals CAS, Rozvojova 1, 165 02 Prague 6, Czech Republic; zdimal@icpf.cas.cz (V.Z.); ondracek@icpf.cas.cz (J.O.); 4Experimental Medicine Centre, Institute for Clinical and Experimental Medicine, Videnska 1958/9, 140 21 Prague 4, Czech Republic; jahb@ikem.cz; 5Department of Machining and Assembly, Department of Engineering Technology, Department of Material Science, Faculty of Mechanical Engineering, Technical University in Liberec, Studentska 1402/2, 461 17 Liberec, Czech Republic; stepanka.dvorackova@tul.cz; 6Department of Nanotoxicology and Molecular Epidemiology, Institute of Experimental Medicine CAS, Videnska 1083, 142 20 Prague 4, Czech Republic; pavel.rossner@iem.cas.cz

**Keywords:** DNA methylation, epigenetics, human, time changes, microarrays, nanoparticles, occupational exposure

## Abstract

A DNA methylation pattern represents an original plan of the function settings of individual cells and tissues. The basic strategies of its development and changes during the human lifetime are known, but the details related to its modification over the years on an individual basis have not yet been studied. Moreover, current evidence shows that environmental exposure could generate changes in DNA methylation settings and, subsequently, the function of genes. In this study, we analyzed the effect of chronic exposure to nanoparticles (NP) in occupationally exposed workers repeatedly sampled in four consecutive years (2016–2019). A detailed methylation pattern analysis of 14 persons (10 exposed and 4 controls) was performed on an individual basis. A microarray-based approach using chips, allowing the assessment of more than 850 K CpG loci, was used. Individual DNA methylation patterns were compared by principal component analysis (PCA). The results show the shift in DNA methylation patterns in individual years in all the exposed and control subjects. The overall range of differences varied between the years in individual persons. The differences between the first and last year of examination (a three-year time period) seem to be consistently greater in the NP-exposed subjects in comparison with the controls. The selected 14 most differently methylated cg loci were relatively stable in the chronically exposed subjects. In summary, the specific type of long-term exposure can contribute to the fixing of relevant epigenetic changes related to a specific environment as, e.g., NP inhalation.

## 1. Introduction

The term “epigenetics”, already coined by the embryologist and developmental biologist Conrad Waddington in the year 1942 [1], was originally defined as ‘‘the branch of biology which studies the causal interactions between genes and their products which bring the phenotype into being’’ [2]. In this millennium, “epigenetics”, including issues related to histone modification, DNA methylation, or miRNA expression, became a phenomenon in various fields of biomedical research. New knowledge on the regulation of DNA function, independent of nucleotides sequence, which is the same for all the cells in an organism, is important and required in countless biological research fields, including genetic toxicology.

Current knowledge identifies two main reasons for DNA methylation and its changes, which generally influence the function of genes. The first one is related to the development of each human, which is generally most dynamic during prenatal development. The second reason is linked with the plasticity of our response to environmental changes, allowing the best survival in new conditions. Numerous combinations of the human DNA methylation pattern are predetermined by roughly 28 million CpG sites that occur at average frequencies of 1/100 bp across the whole genome [3,4]. The methylation/demethylation of cytosine/5-methyl-cytosine, localized in promotor regions, can thus contribute to the switching on and off of the gene expression.

Important knowledge on natural DNA methylation changes over our lifetime, starting from conception, has been accumulated [5]. Current data show that there are three periods in human life during which relatively rapid DNA methylation rearrangements occur and are more susceptible to producing aberrant DNA methylation patterns. They include the period of prenatal development, followed by early childhood, and then the period of older age [6]. Moreover, the methylation pattern changes in certain CpG loci in a particular tissue can predict our biological age, which can differ in two persons of the same age [7,8,9]. Thus, the classification into the category of elderly can be different from an epigenetic point of view in comparison with real age. As already mentioned, the most significant epigenetic changes occur in early development [10,11,12]. Therefore, errors in the methylation process, especially in the earliest phases of pregnancy, caused by adverse environmental conditions acting as teratogens, can be a reason for the generation of an aberrant methylation pattern leading to abortions or developmental defects [13]. In contrast to the fatal effects of some crucial DNA methylation rearrangements, some minor alterations can even be positive for future life and allow adaptation to expected conditions [14]. This phenomenon was described as a predictive adaptive response [15]. In postnatal life, including the periods of childhood, adulthood, and older age, the DNA methylation changes seem to be much slower, or even negligible, in comparison with the prenatal periods [16]. Finally, but importantly, it should be noted that epigenome can also be modified by chronic stressors (described below), which can affect the progress of diseases, such as cancer, diabetes and obesity, infertility, respiratory diseases, allergies, or neurodegenerative disorders [17,18]. Moreover, inside the relatively calm epigenome identified by global methylation methods, many balanced rearrangements (both hypomethylation and hypermethylation) can also occur [13,19].

As already mentioned, numerous studies also show that DNA methylation changes throughout our lifetimes could be affected by our environment, including lifestyle factors and exposures. Thus, in relation to the effect of lifestyle factors, differences were observed, e.g., in newborns of mothers who smoked during pregnancy [20], in children with prenatal alcohol exposure [20], or in adults with prenatal exposure to famine [21]. All these examples relate to the formation of epigenetic memory of events during prenatal development, but some changes in methylation pattern can also be induced in later life, as shown in a study focused on the effects of physical activity [22]. Secondly, concerning environmental exposure episodes, many studies showed the effects of air pollution stressors (particulate matter, polycyclic aromatic hydrocarbons, black carbon, nitrogen oxide, or ozone) across various periods of life by both global (overall hyper/hypo methylation) and genome-wide (loci-specific changes) DNA methylation methods [23,24,25]. Many changes in DNA methylation are also reported in association with occupational exposure to, e.g., pesticides, metals, or NP [19,26,27,28,29].

Despite the published results, there are still many gaps in understanding the dynamics of DNA methylation changes in adulthood, which is generally presented as a relatively stable period during the lifetime. Moreover, data on an individual basis and in the context of more time points are relatively rare. Recently, one study highlighted the importance of mapping both epigenome and exposome and extending the observational timeframe to well before birth [30]. The authors followed for 2 years a group of children with maltreatment in five time points and showed some DNA methylation patterns stable over this timeframe in comparison with the controls. Many authors investigated and confirmed DNA methylation changes on an individual basis after specific interventions, such as weight loss in a timeframe of up to two years [31]. Surprisingly, no data related to longitudinal sampling of exposed adult subjects and DNA methylation profiles have been published.

Thus, in this study, we used our DNA sample collections and chose a total of 14 participants analyzed in our previous NP research [19]. DNA methylation profiling on an individual basis was carried out in four consecutive years (2016–2019) in 10 long-term NP-exposed research workers and 4 nonexposed controls by Infinium Methylation EPIC BeadChips with more than 850,000 CpG loci across the whole human genome. Our aims were to investigate potential changes of the methylome of adults in a relatively short period of time and to determine the possible year-to-year differences and their range on an individual basis in both the exposed and control subjects. This study is a follow-up to our previous work [19], where we focused on the changes in DNA methylation patterns in selected groups; thus, many details have already been published. In this work, we use previous knowledge on differentially methylated sites and, additionally, we investigate individual changes. Therefore, the information on the subgroups is not processed and discussed in detail.

## 2. Results

### 2.1. Basic Characteristics of the Study Subjects

In this study, we focused on individual participants occupationally exposed to NP and nonexposed controls and their DNA methylation changes during a three-year time period. Thus, the data presented in Table 1 include both: (i) data related to groups and (ii) data specific for individual subjects. The selected participants involved in this study were recruited from subjects involved in our sampling performed for four consecutive years (2016–2019), as described in detail in the Materials and Methods section. This data set involved 14 participants (10 occupationally exposed to NP (#1–#10) and 4 nonexposed controls (#11–#14)). No differences between these subgroups were observed for sex (*p* = 0.852), age (*p* = 0.705), and BMI (*p* = 0.380; *p* = 0.407; *p* = 0.492; *p* = 0.396 for individual years, respectively) as well as in previously published characteristics for full groups of 20 exposed and 20 nonexposed control subjects [19]. The age spectrum of the study subjects covered almost the whole range of economically productive adults from 30 to 66 years. The nonexposed controls were slightly older in comparison to the exposed (44 vs. 42 years, respectively), but the NP-exposed group covered a wider age range spectrum. Females were presented in minority (21%; 20% in exposed and 25% in controls), in contrast to 78% males (80% in exposed and 75% in controls). The BMI of all participants covered values from normal weight (minimum 19.2) to obese (maximum 38.9). Individual BMI data were most frequently continuously increasing through the four consecutive years (detailed data per individual years are not shown). The highest increase in BMI was recorded for participant #7 (plus 3.5 between the first and third sampling year). In contrast, the most significant weight loss was observed in participant #6 (BMI decrease 5.1 between the third and fourth sampling year). All the participants were either non-smokers (86%) or weak smokers (14%: #2 and #4). Table 1 also demonstrates the data from questionnaires on the duration of NP exposure. The total exposure includes all occupations linked with possible exposure to NP. In the last column of Table 1, the duration of the most recent occupation connected with similar NP exposure profiles for all nanocomposite research workers (#1–#10) is presented. The data showed a different exposure history for 90% of participants. The longest total exposure was 33 years for participant #6 (whole career), and the longest exposure in the last occupation was 23 years for participant #10. The shortest NP exposure (both total and in the last occupation) was 8 and 7 years, respectively, for participant #5.

### 2.2. Annual Particulate Matter (PM) Monitoring, Including NP Fraction in Sampling Years

According to the questionnaire data, all exposed participants in the study spent part of their work shifts during a sampling day in one of the two NP contaminated workshops (the first one conducting welding and smelting, the second one performing grinding and milling), except the year 2017, when only grinding and milling processes were running. The length of exposure in a sampling day was variable, based on the actual research activities with an average of 60–270 min. The median concentrations of pollutants for individual working activities in four consecutive years (2016, 2017, 2018, and 2019), as well as background data, are summarized in Table 2. Data from an online scanning mobility particle sizer (SMPS) and aerodynamic particle sizer (APS) were used to calculate the median of the number concentration of particulate matter (PM) per cm^3^. Even though the data from 32 size classes/decade were available for this monitoring, we pooled all nanosized fractions smaller than 100 nm, and all fractions larger than 100 nm, into two size fractions as nano PM and major PM. The obtained data show that the risk of exposure differed according to the working process and processed materials. Grinding and milling were generally accompanied by higher production of nano-fraction in comparison with welding, which produces a larger proportion of major PM. In addition, the processed material, as well as the intensity of the work, impacted the final production of the nano-fraction. Specifically, the highest total number concentration of the NP fraction was recorded in 2016, during the grinding and milling of epoxide resin containing nano-SiO_2_. This value was 25× higher in comparison with the year 2019. Annual monitoring also showed a trend of decreasing NP exposure related to both processes (welding and grinding and milling) in sampling days over the years. Interestingly, the ratio of nano PM and major PM proportion changed. The highest proportion of nano PM fraction (96%) was detected in the year 2017 in relation to grinding and milling. Generally, the process of welding was also characterized by high levels of major PM. Overall, the data demonstrate that exposure to NP and major PM fraction changes for individual working processes over the years.

### 2.3. DNA Methylation Profiling During Four Consecutive Years

#### 2.3.1. General Information on the Results

Before comparing the DNA methylation patterns on an individual basis in the exposed and control groups during four consecutive years, we analyzed the proportion of blood cell types in samples from individual years in both groups with the aim to ensure that they do not significantly differ. Potential differences could significantly impact the outcomes of the study. These data are summarized in the Appendix A Figure A1 (for exposed and control subjects). The results of this comparison (B lymphocyte cells, T_h_ helper cells, T_c_ cytotoxic cells, monocytes, neutrophils, natural killer cells; each for years 2016–2019) show generally stable individual blood cell profiles throughout the years in both groups. A summary of blood cell type comparison (*p*-values) for exposed and control subjects in individual years (2016–2019) is presented in the Appendix A Table A1. Although the data show no significant differences, the results for T_c_ cytotoxic cells (CD8T) are consistently (in all years) on the borderline of significance.

We previously analyzed the DNA methylation profiles in a group of NP research workers and controls sampled in 2018 in 80 individual analyses [19]. In this work, a total of 341 CpG loci were significantly hypermethylated and 364 hypomethylated (adjusted *p*-value < 0.05, *p*-value range: 0.0000029–0.049937) when both groups were compared. In addition, the 14 most significant CpG loci with log fold change (FC) > 1.5 and log FC < −1.5 were selected and analyzed in more detail (for an overview of information related to this cg loci, including chromosome, gene and island relation, their relevance and log FC, *p*-values, and adjusted *p*-values see the work of [19]). This follow-up study focused on the individual changes during the three-year time period partly uses these previously obtained results. The results were analyzed in two ways: (i) the whole CpG array data set and (ii) the selected 705 CpG array data set in the group (Figure 1 and Figure 2), as well as on an individual basis including the 14 most significantly affected loci (Figure 3, Figure 4 and Figure 5). The details related to beta values and their changes over time for these 14 loci are presented in Table 3.

#### 2.3.2. The Results on a Group Basis Obtained by PCA

The results of the comparison of the groups analyzed by PCA are presented in Figure 1 and Figure 2. A total of 10 long-term NP-exposed research workers (40 individual array data obtained during four consecutive years) and 4 controls (16 individual array data obtained during four consecutive years) were assessed. The results show no differences in the DNA methylation pattern for the whole CpG array data set (Figure 1), which included more than 850,000 CpG loci per person/year (after subtracting approximately 99,660 CpG loci). In contrast, a trend of separation was found in the first component when the selected 705 CpG array data set was analyzed (Figure 2).

#### 2.3.3. The Results on an Individual Basis Obtained by PCA

The DNA methylation pattern for the whole CpG data set in individual subjects (10 NP-exposed and 4 controls) in four consecutive years is presented in Figure 3. The results show that the DNA methylation pattern tends to change between years for both the exposed and control subjects. The range of changes differs between years and subjects (e.g., the differences between 2017 and 2018 in NP-exposed research workers #3 a #4 were substantially greater in comparison with #2 and #7). Interestingly, the three-year differences (the comparison between the years 2016 and 2019 marked by a black dashed line) were greater in the exposed subjects (approximately 72% on average) in comparison with the controls.

When compared with Figure 3, Figure 4 shows a significantly lower variation (more than 1000-fold) of DNA methylation pattern when 705 CpG loci per subject and year were analyzed. Despite this fact, the dynamics of DNA methylation changes for the exposed and control subjects, as well as in individual years, were also detected. Despite the individual character of changes, the three-year differences (2016 vs. 2019) are substantially smaller in the controls (approximately 60% on average) than in the exposed subjects in this CpG subset.

Similarly, Figure 5 shows the individual DNA methylation pattern trajectories for the 14 most significant CpG loci for both exposed and control subjects. Contrary to the higher differences in the DNA methylation pattern dispersity in the exposed subjects, related to the whole CpG data set and a subset of 705 CpG loci per subject, a relatively stable DNA methylation pattern for these 14 loci was observed in the exposed subjects (approximately 16% lower differences on average) in comparison with the controls.

#### 2.3.4. The Beta Values of the Most Significant CpG loci during the Four Consecutive Years

Following the analysis of the whole CpG array methylome and a subset of the most differently methylated 705 CpG loci by PCA, we assessed in detail the progress of the methylation status in the 14 top, most differently methylated CpG loci between the exposed and control subjects. The level of methylation is presented as beta values (β) related to individual CpG in all four consecutive years. These data, including all descriptive characteristics (mean, standard deviation, median, minimum, and maximum), are presented in Table 3. The results show the distribution of these loci in five genes (*LGR6, RADIL, FGFR2, TMEM9B,* and *FCGBP*), one RNA-encoding gene (*HCG27* *), and one locus, located in the non-coding sequence in chromosome #19. The beta values clearly indicate the differences between the exposed and control subjects (e.g., mean methylation level in the exposed 2019 is approximately 13% higher in cg04811114, approximately 14% higher in cg06825163, approximately 10% lower in cg09622121, etc. when compared with the controls). Moreover, the data suggest higher stability of β values over the three-year period in the exposed subjects than in the controls. This trend is mostly visible in the RNA-encoding gene *HCG27**, where the DNA methylation changes are up to 11% higher (in cg03030317), in comparison with a 1% decrease in this cg in the exposed subjects. Similarly, cg09622121 and cg12771717 in the *HCG27** RNA-encoding gene, as well as cg04811114 and cg06825163 in the LGR6 gene, had a highly stable DNA methylation status over the years in comparison with these CpG in the controls. The trends of DNA methylation changes for these CpG loci for all exposed and control subjects during four consecutive years are shown in Figure 6. Moreover, significant differences (*p* < 0.01) were observed consistently between the groups for these CpG in 2019 (indicated in Table 3).

## 3. Discussion

In our previous study, we found no differences in the DNA methylation pattern following acute NP exposure during one working shift, in contrast to the DNA methylation changes associated with long-term chronic exposure [19]. The dynamics of the DNA methylation changes during the three-year time period in an adult population, with details on an individual basis, was the main goal of this study. Such data have not yet been published, and the mechanisms of individual changes are not well understood. Another advantage of our study is that the methylation profiles of adults, which are presented as stable before menopause in females as well as before the start of the process of aging as a whole [5,6], are generally less studied in comparison with, e.g., newborns. Moreover, the majority of the published studies are still focused on the global methylation changes, which can be hidden by the balanced numbers of hyper- and hypomethylated CpG loci [19,32]. The application of the currently most advanced array chips allows the detection of these balanced differences across the human genome [33].

### 3.1. Exposure Data

Exposure to a vast range of environmental stimuli, including various chemicals and their mixtures or natural radiation background [34] during a lifetime, impacts the methylome that is modified to achieve the best gene expression setting in the context of our environment [35,36,37]. The sum of all exposures over a lifetime, called exposome, is unique for each human. Ten persons involved in our data set were characterized by long-term NP exposure (a total range of 5–33 years for all occupation history, and 4–23 for the last occupation). The exposure levels were measured at the workplace of the participants during all the sampling days (annually, one day in September) in individual years (2016–2019). We presented concentrations of NP as an example; however, the range of exposure can differ during the year/years according to working activities, processed material, distance from exposure sources, and/or intensity of the work. Similarly, NP exposome can also differ from person to person during the years. Our data (Table 2) show, e.g., the highest or even outlying total number of concentrations of nano PM fraction and PM > 100 nm per cm^3^ related to the grinding and milling processes in 2016; this value decreased in other years, however, which agrees with our hypothesis of unique characteristics of every work activity from the exposure point of view.

### 3.2. Overall DNA Methylation Changes in Individual Subjects

Our epigenome is constantly modified by environmental conditions. The changes that protect the organism from the possible adverse effects of harmful pollutants are relatively slow and even negligible when DNA methylation profiles in a short period of time (e.g., a few days) were studied [19,23,37]. Possible moderate DNA methylation changes studied on an individual level during four consecutive years were investigated in this study. As no comparable data have been published, we cannot discuss our results in the context of other studies. Our observations clearly demonstrate the overall DNA methylation changes from year to year in both the exposed and control subjects. Interestingly, the range of these changes differed between the years and the individuals. Generally, the relative stability of DNA methylation fluctuations and possible reversibility of these processes can be demonstrated, e.g., for a comparison of the epigenome between the first and second year of examination when differences were overall more pronounced than between the fourth and first year (Figure 3). This observation can be explained by the continual interaction with the environment, which may differ for individual years. In this study, we focused on NP-exposed subjects, but it is important to consider that each person also has an individual exposome composed of their environmental interactions, including lifestyle factors, that affects the organism during the prevalent part of life spent outside of the working environment. Interestingly, when we compared the fluctuation of the DNA methylation pattern between the exposed and control subjects, we observed higher differences between the overall methylation detected in the first (2016) and last year (2019) in the exposed group. The magnitude of these differences (of overall methylation changes assessed by PCA in Figure 3 and Figure 4) was on average 72% in the case of the full CpG data set and 60% for the selected 705 CpG data set (defined in our previous work [19]), but with more than 1000-fold lower dispersity. This observation agrees with the hypothesis that the epigenome of an exposed subject is increasingly modified due to exposure events that occur over the lifetime, as a result of occupational exposure characterized by the dynamically changing exposure pattern that depends on the dose and processed material.

### 3.3. DNA Methylation Changes Related to the Most Significantly Different CpG Loci

Finally, we focused on the 14 most differently methylated CpG loci, identified in comparison between nanocomposite research workers and control subjects, also investigated in our previous work [19]. Two approaches were used to analyze these loci. The first used the PCA for visualization of the individual DNA methylation pattern trajectories (Figure 5) as well as for the full and selected 705 CpG data set. Contrary to data related to overall DNA methylation pattern changes, a relatively stable DNA methylation pattern for these 14 loci was observed in the exposed subjects with approximately 16% lower differences on average in comparison with the controls. The second approach was based on the comparison of β values of individual CpG in groups of 10 exposed and 4 controls (Table 3) for the individual years. The obtained data demonstrate higher stability in the exposed subjects, particularly for five CpG (Figure 6). These observations indicate that the methylation status of some of the most differently methylated CpG can relate to a possible process of adaptation to the adverse effects of NP exposure and the conservation of this status by means of epigenetic memory [35,36,38]. As previously mentioned, this trend was mostly visible in the RNA gene *HCG27** (HLA complex group 27), where DNA methylation changes were up to 11% higher (for cg03030317) in comparison with a 1% decrease in this cg in the exposed subjects in the three-year period. In addition, in the other two CpG in this gene (cg09622121 and cg12771717), these differences were highly dynamic with the same trends. High differences in % of methylation in the exposed vs. controls in the last sampling year were also observed for cg04811114 (approximately 13% lower β values) and in cg06825163 (approximately 14% lower β values), in both loci in the *LGR6* gene (leucine-rich repeat-containing G). Interestingly, the activity of both genes is associated with the regulation of, e.g., asthma development, lung carcinoma, and signaling pathways that could all be involved in response to exposure stressors [39,40,41]. Thus, we can consider that adaptation of the epigenome to a specific environment can be balanced by the possible negative health effects in later life.

### 3.4. Modification Caused by the Environment

Our previous research focused on methylation profiling by array technology in several population groups (children and adults) characterized by various exposure risks (permanent air pollution and occupational NP) [19,23]. High differences in DNA methylation patterns were found in asthmatic and healthy children (age 7–15 years) from two locations, which substantially differed by air pollution levels. Moreover, unlike the effect of asthma, the impact of permanent exposure to air pollutants was dominantly observed in methylation differences [23]. The second study, which started in the year 2018, investigated the DNA methylation pattern differences in groups of NP-exposed researchers and healthy controls (age 21–72 years). The obtained results showed a substantially lower level of significantly differently methylated CpG loci in comparison with the study comprising children. Moreover, these results were only revealed due to the application of the array methods focusing on loci-specific changes, and allowed to distinguish an almost balanced number of hypomethylated and hypermethylated CpG loci and to identify the differences in the DNA methylation rearrangements in comparison with the global overall hyper/hypomethylation method [19]. All these data also significantly contributed to the formulation of the theory of mechanisms of epigenetic adaptation and epigenetic memory, which can be important for the improved and more effective function of the genes in an adverse environment [35,36]. In this study, the selected and most differently methylated CpG loci were generally highly stable in the chronically exposed group. This observation can be related to the process of adaptation to the exposure to NP, which includes the preservation of the methylation status by epigenetic memory, as supported by previously published hypotheses [35,36,38]. Another interesting mechanism of adaptation to new environmental conditions, as chronic exposure to NP in the occupationally exposed subjects, can also be demonstrated by blood cell type composition. Even though, in our study, significantly different proportions of blood cell types between the exposed and controls were not observed, obtained data showed a consistently higher (on the borderline of significance for four consecutive years) level of Tc cytotoxic cells (CD8T) in the exposed subjects. Interestingly, a similar observation was published two decades ago after long-term occupational exposure to metallic mercury vapors (an increased number of T-cells including CD8T) [42].

### 3.5. Gaps and Challenges for Future Research

Despite the originality of this study, there are still questions and challenges for future research. First, the long-term sampling of the genetic material from the same subjects that would cover a wider period of life and a comparison of differences in the epigenome could significantly contribute to the explanation of the possible adverse effects of various environmental stressors; especially in relation to the development of diseases in later life. An investigation of biological parameters on an individual basis in various periods of life may bring precious data, particularly if they are accompanied by a detailed exposure history, including data from personal monitoring. However, the interpretation of the results may be hampered by the small sample size, as the high variability of the human genome, caused by, e.g., single nucleotide polymorphisms, can complicate the generalization of the findings. Similarly, the sex of subjects may impact the conclusions of the studies. Due to the relatively small number of participants in our study, we only analyzed the DNA methylation pattern changes in autosomes. Focusing on DNA methylation changes through the years in gonosomes could bring new information related to problems with reproduction. The relatively small number of participants also limited the additional detailed statistical analysis, including the calculation of the epigenetic age of participants and the comparison between the exposed and control subjects. Even though we compared these groups and observed a 2.1 year higher epigenetic age in the NP-exposed subjects in comparison with controls (pilot results, data not shown), we suggest focusing on this topic in future studies involving significantly more participants. Another methodological approach could include the sorting of the types of methylated cytosines and the identification of the proportion of 5-methylcytosine (5mC) and 5-hydroxymethylcytosine (5hmC), which can contribute to fast DNA demethylation [43]. Finally, yet importantly, the arrays used in this study cover roughly 3.1% of methylated CpG in the human genome, thus limiting the potential to generalize the data. The application of NGS approaches such as whole epigenome sequencing in combination with whole transcriptome sequencing, and the applications of advanced statistical approaches in the future would allow obtaining more detailed information. Lastly, a proportion of the blood cells should be monitored with the aim to analyze possible differences, especially in large occupationally exposed groups with the goal to clarify the question of the possible effects of exposure. As this study was prevalently focused on individual trajectories in a limited number of subjects and this question could not be credibly answered, this issue remains a challenge for future work.

## 4. Materials and Methods

### 4.1. Study Population Selection

The subjects in this study were selected from the participants of long-term biomonitoring research (conducted annually between years 2015–2020), focused on the chronic and acute effects of occupational exposure to NP in a group of nanocomposite researchers. In contrast to previous research [19] that evaluated the changes in the study groups (pre-shift and post-shift in exposed and controls), this study focused on the changes during the three-year time period on an individual level. Thus, only the subjects sampled repeatedly for consecutive years were included. A total of 14 participants (10 exposed and 4 controls) met the selection criteria (sampling every year between 2016 and 2019). Details of the general characteristics on both group and individual levels (including age, sex, BMI, and length of exposure if applicable) are summarized in Table 1 and described in the results section.

Whole venous blood samples collected from all participants (a total of 56 samples obtained during the three-year time period) were thoroughly mixed with EDTA anticoagulant and kept at 4–10 °C until transportation to the laboratory.

### 4.2. Annual Exposure Monitoring Related to Various Working Activities

Annual stationary monitoring of the risk pollutants was organized in the September of each year between 2016 and 2019 when the biological samples were collected. The monitoring was conducted in the workshops performing the metal active gas welding, smelting, and grinding and milling, where the nanocomposite research workers spend the time during their daily work shifts. Moreover, two background localities were also monitored: (i) in the basement (control for welding and smelting) and (ii) on the ground floor (control for grinding and milling).

Mild steel S355J2 with major Fe content (97.39%) and minor content of Mn, Si, C, P, and S (0.035–1.7%); alloy with AlSi_9_Cu_3_; epoxide resin with up to 20% nano-SiO_2_ filler; and geopolymers with metakaolin, ash or basalt were the main materials processed during annual workshop activities. The processed materials slightly changed during the three-year time period according to actual research activities.

The methods of online exposure monitoring involved the use of two various standard aerosol spectrometers (APS 3321 (Aerodynamic Particle Sizer) and SMPS 3936L (Scanning Mobility Particle Sizer)), both from TSI Inc., St. Paul, MN, USA. Full details related to exposure monitoring techniques included in this long-term study were previously published [44,45,46]. An overview of the stationary monitoring data of nano PM fraction and PM > 100 nm obtained during the annual monitoring of individual working processes is presented in Table 2 and described in the results section.

### 4.3. DNA Methylation Microarray Analysis

The salting-out method described by Miller et al. [47] was used for the DNA extraction from whole venous blood collected into test tubes with EDTA. Aliquots of isolated genomic DNA (gDNA) were stored at −20 °C until the DNA methylation analysis. The quality of DNA, including the concentration, was controlled by the Nanodrop ND-1000 spectrophotometer (Thermo Fischer Scientific, Fitchburg, WI, USA).

A bisulfite conversion for the final recognition of unmethylated cytosine (converted to uracils) and methylated cytosines (remain unchanged) was then carried out. A total of 500 ng gDNA was treated overnight with sodium bisulfite, using Zymo EZ DNA Methylation^TM^ Kit in spin-column format (Zymo Research, Irvine, CA, USA) for this task.

Genome-wide DNA methylation was analyzed by the array-based approach using Qualitative Infinium HD Methylation Assay. MethylationEPIC BeadChips (Infinium) microarray (Illumina, San Diego, CA, USA), allowing the interrogation of more than 850,000 CpG methylation sites dispersed through the whole human genome (all chromosomes and gene and intergenic regions), were used [33]. Individual methodological steps of the analysis of bisulfite-converted gDNA were processed according to the manufacturer’s protocol (Infinium^®^ HD Assay Methylation Protocol Guide #15019519v01 from November 2015, manual version, provided by Illumina). Briefly, the main steps included (i) the enzymatic fragmentation, (ii) precipitation, (iii) resuspension, (iv) overnight hybridization, (v) washing, and (vi) extension and BeadChip staining.

Finally, all the chips were scanned by the iScan System (Illumina, San Diego, CA, USA) for final imaging. The methylation status at each CpG site was estimated by measuring the intensity of the pair of methylated and unmethylated probes. The three random samples were used as replicates on various chips for the quality assurance/quality control assessment (QA/QC). High Pearson’s correlation coefficient was confirmed for these replicates (r = 0.999, *p* < 0.001).

### 4.4. Statistical Approaches

Basic descriptive statistics, including mean, standard deviation (SD), median, minimum, maximum, and t-test for normally distributed variables, were calculated using Microsoft Excel.

Advanced statistical analyses were analyzed using scripts in the R environment and minfi package in Bioconductor [48]. Prior to this analysis, the data were prepared by a series of filtering on methylation probes. A total of 99,660 CpG loci were excluded related to both sex subjects, as analyzed in our data set, because they were below the detection limit due to the non-specificity of probes in single nucleotide polymorphisms sites and to the previously reported cross-reactivity [49]. Moreover, the proportions of individual blood cell types in individual years for both the exposed and the controls, were calculated using package ENmix, which is a Bioconductor tool for the quality control of DNA methylation BeadChips [50].

The principal component analysis (PCA) was the main tool for the comparison of the overall DNA methylation status of all participants in the four consecutive years, 2016–2019, and for analysis of the individual trajectory and range of changes. This tool was used in: (i) full CpG data set obtained from array chips after excluding CpG loci as described; (ii) selected 705 CpG, which were most differently methylated in a group of 20 chronically exposed NP research workers and 20 controls in the year 2018, as mentioned and previously published [19]; and (iii) for the 14 most differentially methylated CpG loci. The range of three-years changes was analyzed from trajectories between the years 2016 and 2019 for both the individual subjects and the exposed and control groups.

Next, DNA methylation levels in the 14 most differently methylated CpGs between the exposed and controls were compared based on the beta (β) values, which estimate the ratio of fluorescent signal intensities between methylated and unmethylated sites. Their values are between 0 and 1 for unmethylated to fully methylated CpG, respectively.

Additionally, trajectories of mean beta values in individual years in selected CpG for the control and exposed groups were shown using ggplot2 (tidyverse package) and generalized by a linear model with smoothed conditional mean.

Pilot testing of epigenetic age was calculated by two methodological approaches [51] (details in Discussion).

## 5. Conclusions

This research focused on questions of the dynamics of the DNA methylation changes over a three-year period during four consecutive years in a group of adult NP-exposed subjects and controls. In contrast to any previous study, the DNA methylation data were also presented on an individual level. Even though the epigenome in adulthood is considered relatively stable in comparison with the next period of life, our data show its slow continual dynamicity, which seems to be higher in the exposed subjects. By comparison, the selected most differently methylated CpG loci were generally highly stable in this chronically exposed group, which could be related to the process of adaptation to the exposure to NP and the fixing of this status by epigenetic memory [33,34,36].

## Figures and Tables

**Figure 1 ijms-22-07834-f001:**
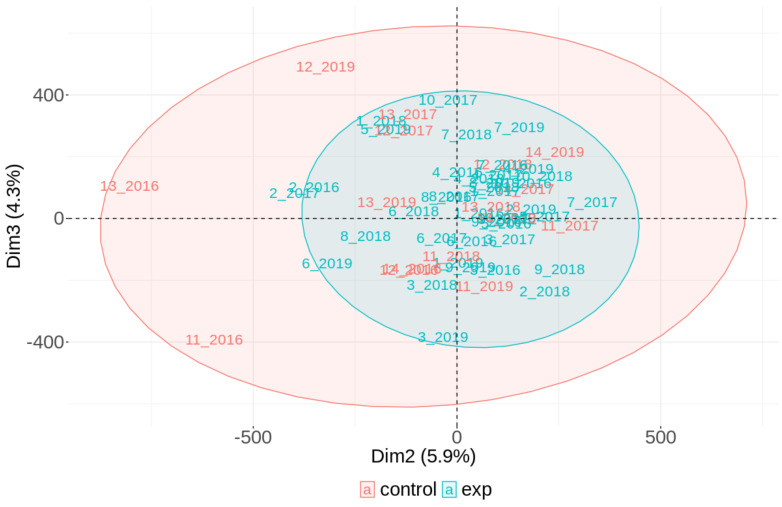
A comparison of the DNA methylation pattern analyzed by PCA in groups of NP-exposed and control subjects pooled from four consecutive years: results for the **whole CpG** data set analyzed by array chips.

**Figure 2 ijms-22-07834-f002:**
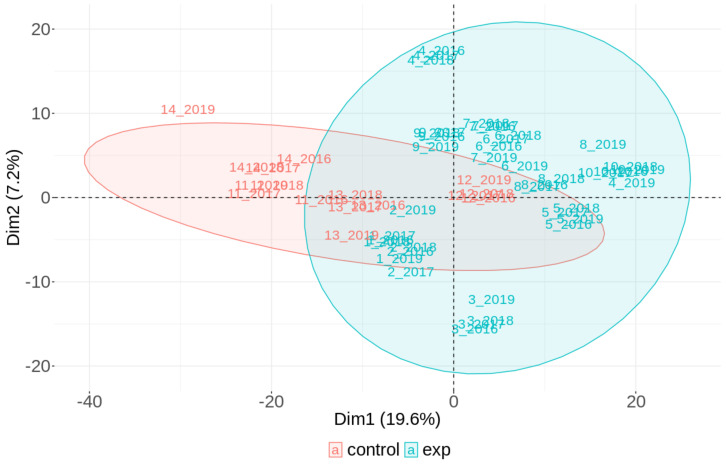
A comparison of the DNA methylation pattern analyzed by PCA in groups of NP-exposed and control subjects pooled from four years: results of the selected **705 CpG** data set analyzed by array chips.

**Figure 3 ijms-22-07834-f003:**
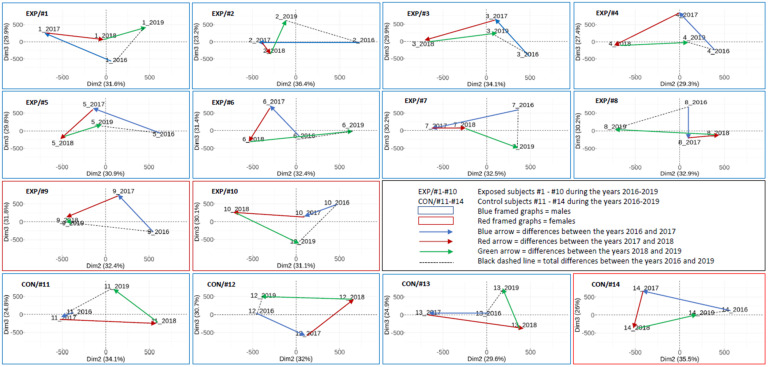
Individual DNA methylation pattern trajectories of the exposed and control subjects analyzed by PCA in four consecutive years: results of the **whole CpG** data set analyzed by array chips. Three-year differences between the years 2016 and 2019 (dashed line) were used for the calculations of the range of changes in the exposed and control subjects (details in Section 2.3.3).

**Figure 4 ijms-22-07834-f004:**
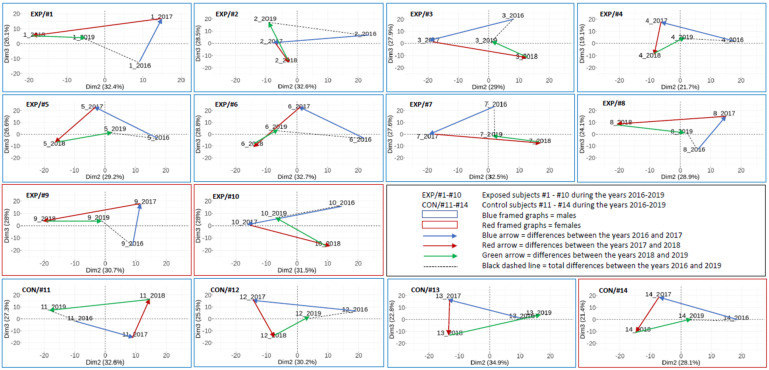
Individual DNA methylation pattern trajectories of the exposed and control subjects analyzed by PCA in four consecutive years: results of the **selected 705 CpG** data set analyzed by array chips. Three-year differences between the years 2016 and 2019 (dashed line) were used for the calculations of the range of changes in the exposed and control subjects (details in Section 2.3.3).

**Figure 5 ijms-22-07834-f005:**
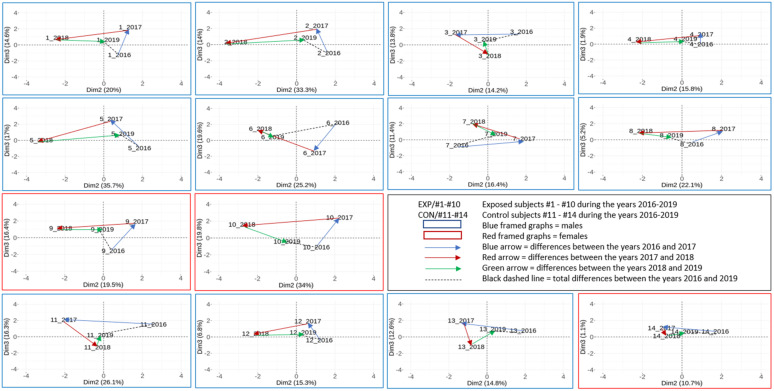
Individual DNA methylation pattern trajectories of the exposed and control subjects analyzed by PCA in four consecutive years: results of the **selected 14 CpG** data set analyzed by array chips. Three-year differences between the years 2016 and 2019 (dashed line) were used for the calculations of the range of changes in the exposed and control subjects (details in Section 2.3.3).

**Figure 6 ijms-22-07834-f006:**
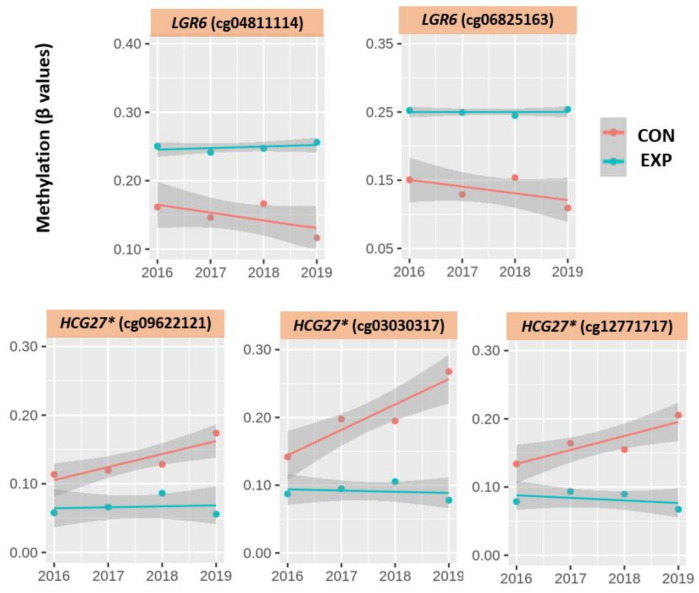
Beta value trajectories of the exposed and control subjects in four consecutive years for 5 CpG loci with significant beta value differences.

**Table 1 ijms-22-07834-t001:** Basic characteristics of the exposed and control subjects on the group and individual levels in four consecutive years.

Group. Participant #	N	Sex ^a^ (M/F)	Age ^a^ (4-Year Range ^b^)	BMI ^a^ (kg/m^2^) (4-Year Range ^c^)	Exposure Total (4-Year Range ^b^)	Exposure of Last Occupation (4-Year Range ^b^)
Exposed (All)	10	8/2	30–66	19.2–36.7	5–33	4–23
#1		M	30–33	24.9–27.8	8–11	7–10
#2		M	31–34	32.9–36.2	6–9	6–9
#3		M	32–35	19.2–21.3	10–13	4–7
#4		M	33–36	25.7–28.8	10–13	7–10
#5		M	36–39	22.7–23.9	5–8	4–7
#6		M	49–52	25.2–30.3	30–33	7–10
#7		M	54–57	33.2–36.7	25–28	9–12
#8		M	63–66	24.8–26.6	10–13	8–11
#9		F	37–40	19.9–21.7	10–13	6–9
#10		F	50–53	26.2–27.9	25–28	20–23
Controls (All)	4	3/1	35–57	20.1–38.9	N/A	N/A
#11		M	39–42	25.6–26.7	N/A	N/A
#12		M	48–51	33.1–35.4	N/A	N/A
#13		M	54–57	36.6–38.9	N/A	N/A
#14		F	35–38	20.1–22.1	N/A	N/A

Abbreviations: N, number of subjects; M, males; F, females; N/A, not applicable; BMI, body mass index; ^a^ (sex, age, BMI): *p* > 0.05 for a comparison between the exposed and control group (details in Section 4.1.); 4-year range ^b^, minimum 2016–maximum in 2019; 4-year range ^c^, minimum total–maximum total.

**Table 2 ijms-22-07834-t002:** Annual monitoring of total number concentration of nano PM fraction and PM > 100 nm per cm^3^ related to individual working processes and background in working places of nanocomposite researchers.

Year	Processing Backgrounds	Nano PM <25–100 nm	Major PM >100–10 µm	Processed Materials
2016	Welding	48,800	72,933	Mild steel S355J2
	Smelting	46,000	2603	AlSi_9_Cu_3_, alloy
	Background in basement	20,100	673	N/A
	Grinding and milling	322,000	204,030	Epoxide resin with nano-SiO_2_
	Background in ground floor	116,000	168,019	N/A
2017	Grinding and milling	89,900	4045	Epoxide resin with nano-SiO_2_ and geopolymers with metakaolin, ash or basalt
	Background in ground floor	4680	4101	N/A
2018	Welding	5520	3791	Mild steel S355J2
	Background in basement	6834	2370	N/A
	Grinding and milling	26,100	3059	Epoxide resin with nano-SiO_2_
	Background in ground floor	3642	1680	N/A
2019	Welding	3793	12,839	Mild steel S355J2
	Background in basement	3017	998	N/A
	Grinding and milling	12,830	1075	Epoxide resin with nano-SiO_2_
	Background in ground floor	1536	728	N/A

Abbreviations: N/A, not applicable.

**Table 3 ijms-22-07834-t003:** DNA methylation level changes during the four consecutive years in the 14 most differently methylated CpG loci between the exposed and control subjects.

	Beta Values in Exposed Subjects #1–#10					Beta Values in Control Subjects #11–#14				
	2016	2017	2018	2019	2016 vs. 2019	2016	2017	2018	2019	2016 vs. 2019
Probe Chr.: *Gene*	Mean ± SD Median (Range)				Mean Diff. (%)	Mean ± SD Median (Range)				Mean Diff. (%)
cg04811114 1: *LGR6*	0.25 ± 0.09 0.26 (0.13–0.37)	0.24 ± 0.09 0.26 (0.13–0.35)	0.24 ± 0.09 0.26 (0.13–0.34)	0.25 ± 0.07 ** 0.27 (0.15–0.36)	0	0.17 ± 0.05 0.15 (0.14–0.24)	0.15 ± 0.01 0.15 (0.14–0.16)	0.17 ± 0.07 0.15 (0.12–0.27)	0.12 ± 0.02 0.13 (0.09–0.14)	−5
cg06825163 1: *LGR6*	0.25 ± 0.09 * 0.28 (0.13–0.35)	0.25 ± 0.10 * 0.27 (0.12–0.37)	0.24 ± 0.09 0.25 (0.13–0.34)	0.25 ± 0.07 ** 0.25 (0.15–0.36)	0	0.16 ± 0.05 0.14 (0.12–0.23)	0.13 ± 0.01 0.13 (0.12–0.15)	0.16 ± 0.05 0.14 (0.13–0.23)	0.11 ± 0.02 0.12 (0.08–0.13)	−4
cg09622121 6: *HCG27* *	0.06 ± 0.03 0.05 (0.03–0.11)	0.07 ± 0.04 0.05 (0.03–0.13)	0.09 ± 0.05 0.07 (0.04–0.17)	0.06 ± 0.02 ** 0.05 (0.03–0.09)	0	0.11 ± 0.02 0.12 (0.08–0.13)	0.11 ± 0.05 0.14 (0.04–0.14)	0.12 ± 0.03 0.13 (0.08–0.15)	0.16 ± 0.08 0.19 (0.05–0.23)	−5
cg03030317 6: *HCG27* *	0.09 ± 0.03 0.08 (0.05–0.16)	0.10 ± 0.04 * 0.08 (0.05–0.19)	0.11 ± 0.05 * 0.10 (0.06–0.19)	0.08 ± 0.02 ** 0.08 (0.06–0.13)	−1	0.14 ± 0.06 0.15 (0.05–0.18)	0.18 ± 0.08 0.21 (0.06–0.25)	0.18 ± 0.08 0.22 (0.07–0.22)	0.25 ± 0.12 0.27 (0.08–0.36)	+11
cg12771717 6: *HCG27* *	0.08 ± 0.05 0.05 (0.03–0.14)	0.09 ± 0.05 0.09 (0.03–0.18)	0.09 ± 0.06 0.06 (0.02–0.24)	0.07 ± 0.03 ** 0.06 (0.04–0.13)	−1	0.13 ± 0.06 0.15 (0.04–0.16)	0.15 ± 0.06 0.17 (0.06–0.21)	0.15 ± 0.06 0.17 (0.05–0.19)	0.19 ± 0.10 0.24 (0.04–0.25)	+6
cg18467790 7: *RADIL*	0.43 ± 0.14 0.42 (0.18–0.69)	0.45 ± 0.15 0.44 (0.21–0.71)	0.44 ± 0.15 0.41 (0.18–0.70)	0.39 ± 0.16 0.35 (0.18–0.79)	−4	0.31 ± 0.08 0.28 (0.25–0.42)	0.32 ± 0.07 0.29 (0.27–0.43)	0.30 ± 0.07 0.28 (0.25–0.41)	0.39 ± 0.08 0.38 (0.33–0.49)	+8
cg07044115 10: *FGFR2*	0.45 ± 0.12 0.39 (0.33–0.64)	0.47 ± 0.12 0.42 (0.32–0.65)	0.46 ± 0.13 0.44 (0.32–0.65)	0.44 ± 0.10 0.41 (0.33–0.65)	−1	0.56 ± 0.10 0.55 (0.47–0.67)	0.54 ± 0.07 0.55 (0.46–0.61)	0.55 ± 0.07 0.54 (0.48–0.62)	0.55 ± 0.17 0.54 (0.40–0.73)	−1
cg16653991 10: *FGFR2*	0.34 ± 0.10 0.34 (0.19–0.47)	0.35 ± 0.09 0.36 (0.19–0.46)	0.33 ± 0.10 0.31 (0.18–0.47)	0.36 ± 0.09 0.39 (0.16–0.45)	+2	0.24 ± 0.07 0.24 (0.16–0.30)	0.26 ± 0.08 0.26 (0.18–0.33)	0.26 ± 0.08 0.26 (0.19–0.34)	0.28 ± 0.16 0.28 (0.11–0.43)	+4
cg10379346 10: *FGFR2*	0.41 ± 0.10 0.41 (0.29–0.58)	0.40 ± 0.11 0.38 (0.26–0.58)	0.41 ± 0.11 0.39 (0.29–0.57)	0.38 ± 0.10 0.36 (0.24–0.61)	−3	0.47 ± 0.10 0.49 (0.35–0.55)	0.48 ± 0.12 0.50 (0.34–0.58)	0.46 ± 0.11 0.48 (0.33–0.56)	0.48 ± 0.16 0.45 (0.34–0.67)	+1
cg06791446 10: *FGFR2*	0.43 ± 0.11 0.46 (0.29–0.60)	0.42 ± 0.10 0.41 (0.30–0.60)	0.45 ± 0.11 0.42 (0.31–0.61)	0.40 ± 0.10 0.37 (0.29–0.63)	−3	0.53 ± 0.09 0.55 (0.40–0.61)	0.53 ± 0.10 0.54 (0.39–0.62)	0.49 ± 0.10 0.51 (0.36 -0.59)	0.52 ± 0.17 0.50 (0.37–0.70)	−1
cg25052156 10: *FGFR2*	0.57 ± 0.10 0.54 (0.43–0.74)	0.58 ± 0.10 0.55 (0.43–0.74)	0.56 ± 0.10 0.52 (0.44–0.72)	0.54 ± 0.09 0.53 (0.43–0.74)	−3	0.65 ± 0.12 0.67 (0.50–0.75)	0.64 ± 0.08 0.67 (0.54–0.71)	0.64 ± 0.08 0.64 (0.56–0.71)	0.62 ± 0.18 0.62 (0.45–0.81)	−3
cg15570860 11: *TMEM9B*	0.77 ± 0.17 0.85 (0.57–0.96)	0.78 ± 0.17 0.86 (0.56–0.97)	0.79 ± 0.18 0.88 (0.57–0.96)	0.79 ± 0.17 0.87 (0.58–0.97)	−2	0.93 ± 0.02 0.92 (0.91–0.95)	0.91 ± 0.05 0.92 (0.84–0.96)	0.91 ± 0.04 0.92 (0.86–0.95)	0.95 ± 0.01 0.95 (0.94–0.97)	+2
cg14764203 19: *FCGBP*	0.52 ± 0.13 0.54 (0.33–0.67)	0.51 ± 0.13 0.54 (0.33–0.70)	0.52 ± 0.13 0.53 (0.33–0.69)	0.53 ± 0.14 0.59 (0.32–0.68)	+1	0.42 ± 0.12 0.42 (0.29–0.55)	0.41 ± 0.10 0.42 (0.30–0.52)	0.40 ± 0.11 0.41 (0.29–0.51)	0.44 ± 0.24 0.45 (0.16–0.68)	+2
cg03635532 19: *FCGBP*	0.63 ± 0.15 0.63 (0.37–0.79)	0.61 ± 0.16 0.56 (0.42–0.79)	0.61 ± 0.15 0.63 (0.39–0.77)	0.65 ± 0.15 0.71 (0.41–0.80)	+2	0.47 ± 0.15 0.46 (0.33–0.63)	0.53 ± 0.14 0.52 (0.41–0.68)	0.49 ± 0.12 0.49 (0.37–0.62)	0.51 ± 0.27 0.55 (0.18–0.77)	+4

Abbreviations: *LGR6,* leucine-rich repeat-containing G; *RADIL,* Rap associating with DIL domain; *FGFR2,* fibroblast growth factor receptor 2; *TMEM9B,* TMEM9 domain family member B; *FCGBP*, Fc fragment of IgG binding protein; *HCG27* *, HLA complex group 27 (* RNA gene); β value comparison between the exposed and control subjects: * *p* < 0.05 and ** *p* < 0.01.

## Data Availability

The data presented in this study are available on request from the corresponding author.

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
