# Peer review of "Individual DNA Methylation Pattern Shifts in Nanoparticles-Exposed Workers Analyzed in Four Consecutive Years"

_ijms, 2021, doi:10.3390/ijms22157834_

Round 1

Reviewer 1 Report

The authors of the manuscript entitled “Individual DNA Methylation pattern shifts in nanoparticles exposed workers analyzed in four consecutive years” address an important question of whether 5meC levels can be influenced by the exposure to nanoparticles in work settings. In this longitudinal follow-up study, they analyze the methylation patterns of 10 exposed individuals and 4 unexposed controls.

The authors claim that exposed subjects present a greater difference in the methylation patterns compared to unexposed between the first and last time point. In addition, Rossnerova et al. also claim that the top 14 differentially methylated CpG loci between the two groups remain constant in the exposed group, while they are changing in the control samples.

The data is clearly presented, but I think that the manuscript is relying too much on principal component analysis. I believe that additional analyses can be performed to improve the manuscript.

1. It is not clear how the differentially methylated cytosines were identified and why only 14 were selected for further analysis. There should be a table of differentially methylated cytosines.

2. As a supplementary file, I suggest plotting side-by-side the cell type distribution for exposed and controls. I would also recommend including a supplementary table with the values for all individuals and time points. Statistical analysis should be performed to test for significant differences among groups/timepoints.

3. The authors should calculate the DNAm age of all the individuals and time points (e.g. http://dnamage.genetics.ucla.edu/) and look if there are differences in the acceleration between groups. 

4. For longitudinal -omics data, there are softwares such as the TCseq package (https://www.bioconductor.org/packages/release/bioc/html/TCseq.html) and/or CoGAPS (https://www.bioconductor.org/packages/release/bioc/html/CoGAPS.html) that can help in the identification of valuable CpG sites. Other software can be used/developed.

5. It would be interesting to show the trajectories of the most variable CpG sites for each individual over the assessed period and how they behave in other individuals. Variable site identification should consider intermediate time points in addition to the first and last values.

6. Minor English revision.

7. I feel the introduction can be shortened and, after a brief discussion of DNA Methylation, it can focus more on previous studies linking DNA methylation and exposure to environmental/external stimuli, especially the ones looking at longitudinal sampling.

Author Response

Comments and Suggestions for Authors

The authors of the manuscript entitled “Individual DNA Methylation pattern shifts in nanoparticles exposed workers analyzed in four consecutive years” address an important question of whether 5meC levels can be influenced by the exposure to nanoparticles in work settings. In this longitudinal follow-up study, they analyze the methylation patterns of 10 exposed individuals and 4 unexposed controls.

The authors claim that exposed subjects present a greater difference in the methylation patterns compared to unexposed between the first and last time point. In addition, Rossnerova et al. also claim that the top 14 differentially methylated CpG loci between the two groups remain constant in the exposed group, while they are changing in the control samples.

The data is clearly presented, but I think that the manuscript is relying too much on principal component analysis. I believe that additional analyses can be performed to improve the manuscript.

Re.: First, we would like to thank the reviewer for the valuable comments, which helped to improve this manuscript. We absolutely agree with the comment, that the PCA was a preferred statistical tool in this study. Because the investigation of trajectories of the DNA methylation patterns on the individual basis was the main aim of this work, we chose this analysis as a most suitable for the visualization of the results. Some improvements and additional analyses were done in the revised version of the manuscript based on the suggestions and comments below.

  1. It is not clear how the differentially methylated cytosines were identified and why only 14 were selected for further analysis. There should be a table of differentially methylated cytosines.

Re1.: A total of 705 differentially methylated cytosines between the exposed subjects and the controls (adjusted p value < 0.05, range: 0.0000029 – 0.049937) as well as 14 most significant results with log fold change (FC) > 1.5 and log FC < -1.5 were identified in our previous work [1]. This explanation is in the text, section 2.2.1. (General information on the results). Some details were added into the text. The list of these 14 cg loci including the relevant genes and chromosomes is in the Table 3. More details related to the individual cg (island relation, relevance or phenotype, log FC, p-value, p-value adjusted) were already published (Table 2 of the previous work) [1]. We also mentioned this fact in the text.

  1. As a supplementary file, I suggest plotting side-by-side the cell type distribution for exposed and controls. I would also recommend including a supplementary table with the values for all individuals and time points. Statistical analysis should be performed to test for significant differences among groups/timepoints.

Re2.: As suggested, the data are now presented in an appendix (Figure A1) Moreover, a supplementary table of p values (Blood cell type comparisons of exposed and control subjects for individual years (2016-2019)) was also added as Table A1 into Appendix. Uniformly, there are no significant differences as mentioned in the original text, although data for CD8T: Tc cytotoxic cells were on the borderline of significance. We mentioned this fact in discussion along with a new citation [2].

  1. The authors should calculate the DNAm age of all the individuals and time points (e.g. http://dnamage.genetics.ucla.edu/) and look if there are differences in the acceleration between groups.

Re3.: We agree with this interesting suggestion because we had a similar idea as we mentioned the aspect of epigenetic age in the introduction. We have already calculated the DNAm age by two approaches (by Horvath and by Hannum epigenetic clocks) and obtained very interesting results which indicate about 2.1 year higher epigenetic age in the NPs exposed group in comparison with the controls (identical results by both approaches). We now mention this preliminary result in discussion and as a challenge for future research. In statistical approaches, a citation was added [3]. As the cohorts are relatively small (10 exposed vs. 4 controls), we in fact aimed to prepare a separate manuscript concerning this topic as we have additional data from various exposure studies which allow evaluation of these effects more properly.

  1. For longitudinal -omics data, there are softwares such as the TCseq package (https://www.bioconductor.org/packages/release/bioc/html/TCseq.html) and/or CoGAPS (https://www.bioconductor.org/packages/release/bioc/html/CoGAPS.html) that can help in the identification of valuable CpG sites. Other software can be used/developed.

Re4.: Thank you for this valuable recommendation. In this manuscript, and in our previous study [1] we used minfi package in Bioconductor because it is explicitly suggested to analyze Illumina Infinium DNA methylation arrays data. Moreover, ENmix package Bioconductor tool was used for analysis of individual blood cells types. Related to the TCseq package software, which is recommended especially for „time course sequencing data analysis”, we plan to utilize this approach in our next study focused on whole epigenome sequencing. Because this manuscript is a follow-up of our previously published work [1], we used the same approaches, although reviewer’s suggestions are highly valuable for our future work. In this manuscript, detail identification of 14 most differently methylated CpG loci between the exposed and control subjects is presented in Table 3, now including the significance level. Additionally, we prepared new Figure 6 (Beta values trajectories in groups of exposed and control subjects in four consecutive years for 5 CpG loci with significant beta values differences) with the aim to visualize most important data. The section 4.4. (Statistical Approaches) was modified.

  1. It would be interesting to show the trajectories of the most variable CpG sites for each individual over the assessed period and how they behave in other individuals. Variable site identification should consider intermediate time points in addition to the first and last values.

Re5.: New Figure 5 showing individual trajectories of 14 most differently methylated CpG was added including some details suggested by Reviewer 2. Figure legends for Figures 3-5 were modified and details on results generation were included. Some details were also added into the sections 2.2.1, 2.2.3. and into the section „ Statistical approaches “. A typo in % of individual DNA methylation pattern differences comparison was corrected (previously: 60% higher dispersity for whole CpG set in exposed – 62% higher dispersity for 705 CpG subset in exposed was corrected on: 72% higher dispersity for whole CpG set in exposed – 60% higher dispersity for 705 CpG subset in exposed). Also, new information about 16% lower dispersity for 14 CpG subset in the exposed group analyzed based on new PCA analysis (Figure 5) supporting our conclusion was added.

  1. Minor English revision.

Re6.: Some language editing was done based on the suggestion of Reviewer 2. Additionally, the manuscript was again editing by native speaker.

  1. I feel the introduction can be shortened and, after a brief discussion of DNA Methylation, it can focus more on previous studies linking DNA methylation and exposure to environmental/external stimuli, especially the ones looking at longitudinal sampling.

Re7.: As suggested, we shortened the first part of introduction and expanded a part related to DNA methylation and exposure with the special interest related to longitudinal sampling and DNA methylation trajectories. Our original statement: „data on an individual basis have not yet been published“ was corrected to correspond to a new finding Although we originally aimed to underline in introduction a both aspects of epigenetic changes during the life (natural with age and environmentally stimulated), we hope, that now both parts are more balanced. Additionally, new citations were added [4,5].

References

[1] Rossnerova, A.; Honkova, K.; Pelclova, D.; Zdimal, V.; Hubacek, J.A.; Chvojkova, I.; Vrbova, K.; Rossner, P., Jr.; Topinka, J.; Vlckova, S.; Fenclova, Z.; Lischkova, L.; Klusackova, P.; Schwarz, J.; Ondracek, J.; Ondrackova, L.; Kostejn, M.; Klema, J.; Dvorackova, S. DNA methylation profiles in a group of workers occupationally exposed to nanoparticles. Int. J. Mol. Sci. 2020, 21, 1-20.

[2] Moszcyński, P.; Rutowski, J.; Słowiński, S.; Bern, S.; Jakus-Stoga, D. Effects of occupational exposure to mercury vapors on T-cell and NK-cell populations. Arch. Med. Res. Winter 1996, 27, 503-507.

[3] Armstrong, N.; Mather, K.A.; Thalamuthu, A.; Wright, M.J.; Trollor, J.N.; Ames, D.; Brodaty, H.; Schofield, P.R.; Sachdev, P.S.; Kwok, J.B. Aging, exceptional longevity and comparisons of the Hannum and Horvath epigenetic clocks. Epigenomics 2017, 9, 689-700.

[4] Martins, J.; Czamara, D.; Sauer, S.; Rex-Haffner, M.; Dittrich, K.; Dörr, P.; de Punder, K.; Overfeld, J.; Knop, A.; Dammering, F.; Entringer, S.; Winter, S.M.; Buss, C.; Heim, Ch. Childhood adversity correlates with stable changes in DNA methylation trajectories in children and converges with epigenetic signatures of prenatal stress. Neurobiol. Stress 2021, 15, 1-13.

[5] Aronica, L.; Levine, A.J.; Brennan, K.; Mi, J; Gardner, Ch.; Haile, R.W.; Hitchins, M.P. A systematic review of studies of DNA methylation in the context of a weight loss intervention. Epigenomics 2017, 9, 769-787.

Reviewer 2 Report

As it stands the manuscript describes the DNA methylation profiling of bloods cells from workers occupationally exposed to NP over a 3-year period. Some attempt has been made to measure individual workers exposures over this time period and it is recognised that exposure is varied both between individuals and across years. But there is no information on whether individuals took part in all processes, worked the same number of hours, or why two processes are missing from 2017. Is it possible to estimate individual exposures over the 3-year period? It would be interesting to investigate if there was any correlation between levels of exposure and methylation changes (group or individual) – ie was there greater changes in methylation in the years/individuals with the greatest exposures? This would provide greater evidence that the changes were due to the NP exposure.

Throughout the analysis it is unclear what or if any statistical analysis has been performed on differences between exposed and control groups. The authors describe changes – eg in BMI, in DNA methylation, and in beta-values but do not provide any idea of whether these differences are statistically significant (no p values or description of tests done are provided). There is also no comparison of the beta values between exposed workers vs control subjects. Thus, it is difficult to assess whether or not there are any exposure-induced changes.

Finally, there is no real discussion on the potential biological relevance of the identified changes. In the introduction, the authors talk about the importance of DNA methylation for switching genes on and off, and yet they do not discuss even the potential effect of the DNA methylation changes they identify on gene transcription, let alone validate it in any way. This significantly reduces the impact of the work.

Minor revisions/typos:

The manuscript is generally well-written, with good English. However, there are a few odd sentences that need clarification, places where ‘the’ needs to be removed or added, and words where alternatives would be better. I have noted when I have found these – but it would be worth proof-reading again to check.

Line 26 ‘generate the changes’ – remove the

Line 28 ‘in occupationally exposed workers repeatedly, sampled over’ – needs revision – do the authors mean workers that were repeatedly exposed?

Lines 29-31 – could combine these two sentences – a bit repetitive

Line 31 ‘The microarray-based approach’ – change The to A

Line 33 ‘by the Principal’ – remove the

Line 39 ‘to the fixing of relevant epigenetic changes related to’ – relevant to what? Do the authors means ‘to the fixing of epigenetic changes related to’?

Line 41 – add occupational to key words?

Line 43 ‘by an embryologist’ – change an to the

Line 51 ‘nucleotides sequence which’ add comma after sequence

Line 62 ‘the on and off switching of the gene expression’ change to ‘the switching on and off of gene expression’

Line 63 ‘over the lifetime’ change to ‘over our lifetimes’

Lines 71-73 – not really sure what is meant by this sentence and why part of it is underlined? Are the authors trying to say that the most significant epigenetic changes occur in early development?

Lines 75-77 - Might want to clarify this a bit - say why there is a second demethylation - and that only occurs in germ cells?

Line 94 – ‘by chronical stressors’ change chronical to chronic

Line 101 ‘throughout the lifetime’ – change to throughout our lifetimes or throughout and individual’s lifetime

Line 102 plus others ‘exposure episodes’ should just be called exposures

Lines 105-107 – sentence confusing – do the authors mean ‘All these examples relate to the formation of an epigenetic memory of events during prenatal development, but some changes in methylation pattern can also be induced in later life as shown in a study focused on the effects of physical activity [22].’?

Line 110 - what is meant by global vs genome-wide? global changes ie overall hyper/hypo methylation vs loci-specific changes? Need to clarify.

Line 111 ‘this epigenetic marker’ change to DNA methylation

Line 117 - above-described? The NP research of the authors has not been described above? More detailed description of it should go in the introduction to help explain the background to the current study.

Line 126 ‘we individually focused on participants’ change to we focused on individual participants’

Within results – when quoting numbers, if use 1dp for one, then should quote all numbers to 1dp

Line 138 ‘The BMI of all participants in the individual years’ – remove in the individual years

Throughout the authors use ‘prevalently’ – remove this word as it doesn’t add anything

Table 1 – what do * and ** signify?

Line 197 ‘As already mentioned’ – has not already been mentioned here - so remove and just begin with 'We previously analyzed...'. Also, some info about this previous study should go in the introduction.

Line 200 ‘addition, 14 most’ change to ‘addition, the 14 most’

Line 233 change dispersity to variation

Line 239 and others ‘during the Years’ is used – should be changed throughout to during the 4-year time period

Line 249 – change about to approximately – here and throughout

Line 255 and others change cgs to CpGs?

Figure legends - More detail on how results generated - for all figures

Line 282 – defection should be detection

Lines 290-295 – sentences needs clarification – not sure what is meant

Lines 341-344 – refs for this?

Lines 355-358 – sentence needs clarification – do the authors mean that it is important to look at loci-specific changes and not just overall global hyper/hypo methylation?

Some parts of the discussion are repetitive – it could be made more concise – perhaps the use of subheading would help?

Line 466 ‘levels in 14 most’ add the before 14 and add s to CpG later in the sentence

In materials and methods it would be nice to include some extra detail instead of just saying was performed as described previously – the reader doesn’t necessarily want to have to look up multiple papers to work out how the experiments were done.

Author Response

Comments and Suggestions for Authors

As it stands the manuscript describes the DNA methylation profiling of bloods cells from workers occupationally exposed to NP over a 3-year period. Some attempt has been made to measure individual workers exposures over this time period and it is recognised that exposure is varied both between individuals and across years.

Re.: We thank the reviewer for the comments and suggestions. We also thank for careful reading of the text and typos identification. Additional language editing was performed by native speaker.

But there is no information on whether individuals took part in all processes, worked the same number of hours, or why two processes are missing from 2017. Is it possible to estimate individual exposures over the 3-year period? It would be interesting to investigate if there was any correlation between levels of exposure and methylation changes (group or individual) – ie was there greater changes in methylation in the years/individuals with the greatest exposures? This would provide greater evidence that the changes were due to the NP exposure.

Re.: We added some details into the section 2.2. The time range (different for individual participants) is mentioned. Additional information related to presence in one of the two workshops is also included in text, as along with the explanation why two processes in 2017 are missing. Regarding the estimation of individual exposure over the 3-year period, we do not have personal data in this study. Moreover, exposure doses are highly different depending on processed materials in individual years, intensity of the work and/or distance of the source of exposure or from stationary monitor. Thus, we have not done this evaluation, but we plan to perform it in a cohort sampled in year 2020 for which personal monitors (PENS) were used. Therefore, we cannot correlate the levels of exposure and methylation changes. We mentioned personal monitors in discussion in the part concerning challenges and gaps (3.5.).

Throughout the analysis it is unclear what or if any statistical analysis has been performed on differences between exposed and control groups. The authors describe changes – eg in BMI, in DNA methylation, and in beta-values but do not provide any idea of whether these differences are statistically significant (no p values or description of tests done are provided). There is also no comparison of the beta values between exposed workers vs control subjects. Thus, it is difficult to assess whether or not there are any exposure-induced changes.

Re.: Basic statistical analysis was performed for comparison of sub-groups (10 exposed and 4 controls) for general characteristics (gender, age, BMI). Information was added to Table 1, and some details (p-values) were added to section 2.1. These trends are identical as in our previous work [1] that focused exclusively on the comparison of the groups, in contrast to this manuscript, in which details on individual trajectory in DNA methylation were investigated. In addition, beta values comparisons (significant differences) between the exposed and control subjects for individual CpG were added to Table 3. Moreover, new Figure 6 (Beta values trajectories in groups of exposed and control subjects in four consecutive years for 5 CpG loci with significant beta values differences) was added.

Finally, there is no real discussion on the potential biological relevance of the identified changes. In the introduction, the authors talk about the importance of DNA methylation for switching genes on and off, and yet they do not discuss even the potential effect of the DNA methylation changes they identify on gene transcription, let alone validate it in any way. This significantly reduces the impact of the work.

Re.: The discussion was extensively modified and split by subheadings into individual topics. Moreover, three new references were added that focus on the most significantly differentially methylated genes (CpG) for which greatest changes were observed during the four consecutive years [2-4]. Also, stability of the methylation level in the exposed subjects in selected CpG is now discussed throughout the text and explained by the epigenetic adaptation including its preservation by epigenetic memory and potential negative biological relevance in later life.

We are aware of the profit of transcriptomics data. Unfortunately, we have no biological material for RNA extraction from samples collected in the 2016-2019 period. The material (whole venous blood) was only collected in September 2020 sampling and a new grant application that includes this topic was already submitted. We mentioned the need of transcriptomic data in the challenges for future research in the discussion.

References

[1] Rossnerova, A.; Honkova, K.; Pelclova, D.; Zdimal, V.; Hubacek, J.A.; Chvojkova, I.; Vrbova, K.; Rossner, P., Jr.; Topinka, J.; Vlckova, S.; Fenclova, Z.; Lischkova, L.; Klusackova, P.; Schwarz, J.; Ondracek, J.; Ondrackova, L.; Kostejn, M.; Klema, J.; Dvorackova, S. DNA methylation profiles in a group of workers occupationally exposed to nanoparticles. Int. J. Mol. Sci. 2020, 21, 1-20.

[2] Hoang, T.T.; Sikdar, S.; Xu, Ch.-J.; Lee, M.K.; Cardwell, J.; Forno, E.; Imboden, M.; Jeong, A.; Madore, A.-M.; Qi, C.; Wang, T.; Bennett, B.D.; Ward, J.M.; Parks, Ch.G.; Beane-Freeman, L.E.; King, D.; Motsinger-Reif, A.; Umbach, D.M.; Wyss, A.B.; Schwartz, D.A.; Celedón, J.C.; Laprise, C.; Ober, C.; Probst-Hensch, N.; Yang, I.V.; Koppelman, G.H.; London, S.J. Epigenome-wide association study of DNA methylation and adult asthma in the Agricultural Lung Health Study. Eur. Respir. J. 2020, 56, 1-187.

[3] Cortesi, E.; Ventura, J.J. Lgr6: From Stemness to Cancer Progression. J. Lung Health Dis. 2019, 3, 12-15.

[4] Raslan, A.A.; Yoon, J.K. R-spondins: Multi-mode WNT signaling regulators in adult stem cells. Int. J. Biochem. Cell Biol. 2019, 106, 26-34.

Minor revisions/typos:

The manuscript is generally well-written, with good English. However, there are a few odd sentences that need clarification, places where ‘the’ needs to be removed or added, and words where alternatives would be better. I have noted when I have found these – but it would be worth proof-reading again to check.

Line 26 ‘generate the changes’ – remove the

Re.: Removed as suggested.

Line 28 ‘in occupationally exposed workers repeatedly, sampled over’ – needs revision – do the authors mean workers that were repeatedly exposed?

Re.: The sentence was revised.

Lines 29-31 – could combine these two sentences – a bit repetitive

Re.: Repetitive information was removed.

Line 31 ‘The microarray-based approach’ – change The to A

Re.: Changed as suggested.

Line 33 ‘by the Principal’ – remove the

Re.: Removed as suggested.

Line 39 ‘to the fixing of relevant epigenetic changes related to’ – relevant to what? Do the authors means ‘to the fixing of epigenetic changes related to’?

Re.: The world „relevant“ was excluded.

Line 41 – add occupational to key words?

Re.: Keywords were modified.

Line 43 ‘by an embryologist’ – change an to the

Re.: Changed as suggested.

Line 51 ‘nucleotides sequence which’ add comma after sequence

Re.: Comma was added.

Line 62 ‘the on and off switching of the gene expression’ change to ‘the switching on and off of gene expression’

Re.: Changed as suggested.

Line 63 ‘over the lifetime’ change to ‘over our lifetimes’

Re.: Changed as suggested.

Lines 71-73 – not really sure what is meant by this sentence and why part of it is underlined? Are the authors trying to say that the most significant epigenetic changes occur in early development?

Re.: The sentenced was underlined by mistake. The text was simplified according to the suggestion.

Lines 75-77 - Might want to clarify this a bit - say why there is a second demethylation - and that only occurs in germ cells?

Re.: This part of the text was completely removed based on the request of reviewer 1.

Line 94 – ‘by chronical stressors’ change chronical to chronic

Re.: hanged as suggested.

Line 101 ‘throughout the lifetime’ – change to throughout our lifetimes or throughout and individual’s lifetime

Re.: Changed to „ throughout our lifetimes “.

Line 102 plus others ‘exposure episodes’ should just be called exposures

Re.: The text was modified.

Lines 105-107 – sentence confusing – do the authors mean ‘All these examples relate to the formation of an epigenetic memory of events during prenatal development, but some changes in methylation pattern can also be induced in later life as shown in a study focused on the effects of physical activity [22].’?

Re.: The sentence was clarified as suggested.

Line 110 - what is meant by global vs genome-wide? global changes ie overall hyper/hypo methylation vs loci-specific changes? Need to clarify.

Re.: The text was clarified as suggested.

Line 111 ‘this epigenetic marker’ change to DNA methylation

Re.: The text was changed.

Line 117 - above-described? The NP research of the authors has not been described above? More detailed description of it should go in the introduction to help explain the background to the current study.

Re.: The text was clarified with the aim to highlight the main intention of this manuscript (individual DNA methylation changes).

Line 126 ‘we individually focused on participants’ change to we focused on individual participants’

Re.: The text was modified.

Within results – when quoting numbers, if use 1dp for one, then should quote all numbers to 1dp

Re.: The numbers were modified.

Line 138 ‘The BMI of all participants in the individual years’ – remove in the individual years

Re.: The text was removed.

Throughout the authors use ‘prevalently’ – remove this word as it doesn’t add anything

Re.: We identified this word at two places in the text. The first of them we replaced by „ most frequently “and the second was removed.

Table 1 – what do * and ** signify?

Re.: No significant differences between the groups was observed. Asterisks and legend were modified.

Line 197 ‘As already mentioned’ – has not already been mentioned here - so remove and just begin with 'We previously analyzed...'. Also, some info about this previous study should go in the introduction.

Re.: The text was modified, including introduction.

Line 200 ‘addition, 14 most’ change to ‘addition, the 14 most’

Re.: The text was changed.

Line 233 change dispersity to variation

Re.: The text was changed.

Line 239 and others ‘during the Years’ is used – should be changed throughout to during the 4-year time period

Re.: We revised the text and now use either: “in four consecutive years” (2016, 2017, 2018, 2019), or: “during the three-year time period” especially when we compare the changes between the years 2016 and 2019.

Line 249 – change about to approximately – here and throughout

Re.: We changed it through the text.

Line 255 and others change cgs to CpGs?

Re.: The text was changed.

Figure legends - More detail on how results generated - for all figures

Re.: New Figure 5 related to individual trajectories of 14 most differently methylated CpG was added as suggested by Reviewer 1. Figure legends for Figures 3-5 were clarified and details were added. More details are also provided in sections 2.2.1, 2.2.3. and „ Statistical approaches “. A typo of % of individual DNA methylation pattern differences comparison was corrected (previously: 60% higher dispersity for whole CpG set in exposed – 62% higher dispersity for 705 CpG subset in exposed was corrected on: 72% higher dispersity for whole CpG set in exposed – 60% higher dispersity for 705 CpG subset in exposed). Also, new information about 16% lower dispersity for 14 CpG subset in the exposed group based on new PCA analysis (Figure 5) supporting our conclusion was added.

Line 282 – defection should be detection

Re.: The text was corrected.

Lines 290-295 – sentences needs clarification – not sure what is meant

Re.: The text was modified and clarified.

Lines 341-344 – refs for this?

Re.: Three references were added.

Lines 355-358 – sentence needs clarification – do the authors mean that it is important to look at loci-specific changes and not just overall global hyper/hypo methylation?

Re.: The text was clarified according to suggestion.

Some parts of the discussion are repetitive – it could be made more concise – perhaps the use of subheading would help?

Re.: The discussion was modified. Some information was excluded, and some added, based on the comments of both reviewers. Moreover, new structure with subheading was implemented as suggested 3.1. – 3.5.

Line 466 ‘levels in 14 most’ add the before 14 and add s to CpG later in the sentence

Re.: The text was corrected.

In materials and methods it would be nice to include some extra detail instead of just saying was performed as described previously – the reader doesn’t necessarily want to have to look up multiple papers to work out how the experiments were done.

Re.: Some modifications were done in this section. However, preparation of the samples for DNA methylation analysis includes numerous methodological steps (more than 100 pages in the Illumina instruction manual). Thus, we mentioned basic steps only with the identification of the original protocol (available on the web): „Individual methodological steps of the analysis of bisulfite-converted gDNA were processed according to manual version of the manufacturer’s protocol (Infinium® HD Assay Methylation Protocol Guide #15019519v01 from November 2015, provided by Illumina). “

Round 2

Reviewer 1 Report

The authors satisfactorily addressed the reviewer's comments.